# Two-scale multi-model ensemble:
# Is a hybrid ensemble of opportunity telling us more?

Stefano Galmarini[1], Ioannis Kioutsioukis[18], Efisio Solazzo[1], Ummugulsum Alyuz[6] , Alessandra Balzarini[7], Roberto Bellasio[2], Anna M. K. Benedictow[22], Roberto Bianconi[2], Johannes Bieser[9], Joergen Brandt[10], Jens H.Christensen[10], Augustin Colette[11], Gabriele Curci[4,5,] , Yanko Davila[22], Xinyi Dong[20], Johannes Flemming[19], Xavier Francis[12], Andrea Fraser[13], Joshua Fu[20], Daven Henze[21], Christian Hogrefe[3], Ulas Im[10], Marta Garcia Vivanco[14], Pedro Jiménez-Guerrero[8],  Jan Eiof Jonson[22], Nutthida Kitwiroon [16], Astrid Manders[15], Rohit Mathur[3], Laura Palacios-Peña[8], Guido Pirovano[7], Luca Pozzoli[6,1], Marie Prank[17], Martin Schultz[22], Rajeet S. Sokhi[12], Kengo Sudo[24], Paolo Tuccella[5], Toshihiko Takemura[23], Takashi Sekiya[24], Alper Unal[6]

European Commission, Joint Research Centre, JRC, Ispra (VA), Italy
Enviroware srl, Concorezzo, MB, Italy
Computational Exposure Division - NERL, ORD, U.S. EPA
CETEMPS, University of L'Aquila, Italy
Dept. Physical and Chemical Sciences, University of L'Aquila, Italy
Eurasia Institute of Earth Sciences, Istanbul Technical University, Turkey
Ricerca sul Sistema Energetico (RSE SpA), Milano, Italy
University of Murcia, Department of Physics, Physics of the Earth, Facultad de Química, Campus de Espinardo, 30100 Murcia, Spain
Institute of Coastal Research, Chemistry Transport Modelling Group, Helmholtz-Zentrum Geesthacht, Germany
Aarhus University, Department of Environmental Science, Frederiksborgvej 399, 4000 Roskilde, Denmark
INERIS, Institut National de l'Environnement Industriel et des Risques, Parc Alata, 60550 Verneuil-en-Halatte, France
Centre for Atmospheric and Instrumentation Research (CAIR), University of Hertfordshire, Hatfield, UK
Ricardo Energy & Environment, Gemini Building, Fermi Avenue, Harwell, Oxon, OX11 0QR, UK
CIEMAT. Avda. Complutense, 40.  28040. Madrid, Spain
Netherlands Organization for Applied Scientific Research (TNO), Utrecht, The Netherlands
Environmental Research Group, Kings' College London, London, United Kingdom
Finnish Meteorological Institute, Atmospheric Composition Research Unit, Helsinki, Finland
University of Patras, Physics Department, Laboratory of Atmospheric Physics, 26504 Rio, Greece
European Centre for Medium-Range Weather Forecasts, Reading, UK
Department of Civil and Environmental Engineering, The University of Tennessee, Knoxville, TN, 37919, USA
Department of Mechanical Engineering, University of Colorado, 1111 Engineering Drive, Boulder, CO, USA.
Norwegian Meteorological Institute, Oslo, Norway
Research Institute for Applied Mechanics, Kyushu University, Fukuoka, Japan
Japan Agency for Marine-Earth Science and Technology, Yokohama, Japan

**Abstract**

In this study we introduce a *hybrid ensem*ble consisting of air quality models operating at both the global and regional scale. The work is motivated by the fact that these different types of models treat specific portions of the atmospheric spectrum with different levels of detail and it is hypothesized that their combination can generate an ensemble that performs better than mono-scale ensembles. A detailed analysis of the hybrid ensemble is carried out in the attempt to investigate this hypothesis and determine the real benefit it produces compared to ensembles constructed from only global scale or only regional scale models. The study utilizes 13 regional and 7 global models participating in the HTAP2/AQMEII3 activity and focuses on surface ozone concentrations over Europe for the year 2010. Observations from 405 monitoring rural stations are used for the evaluation of the ensemble performance. The analysis first compares the modelled and measured power spectra of all models and then assesses the properties of the mono-scale ensembles, particularly their level of redundancy, in order to inform the process of constructing the hybrid ensemble. This study has been conducted in the attempt to identify that the improvements obtained by the hybrid ensemble relative to the mono-scale ensembles can be attributed to its hybrid nature. The improvements are visible in a slight increase of the diversity and a marked improvement of the accuracy. The results show that the optimal set is constructed from an equal number of global and regional models at only 15% of the stations. This implies that for the majority of the cases the regional scale set of models governs the ensemble. However given high degree of redundancy that characterises the regional scale models, no further improvement could be expected in the ensemble performance by adding yet more regional models to it. Therefore the improvement obtained with the hybrid set can confidently be attributed to the different nature of the global models. The study strongly reaffirms the importance of an in-depth inspection of any ensemble of opportunity in order to extract the maximum amount of information and to have full control over the data used in the construction of the ensemble.

## 1. Introduction

It has been widely demonstrated (e.g Potempsky and Galmarini, 2009) that when multiple model results are distilled to retain only original and independent contributions (Solazzo et al. 2012) and thereafter statistically combined in what is usually called an ensemble, one obtains results that are systematically superior to the performance of the individual models and therefore can provide more accurate and robust assessments or predictions.

An additional advantage of using an ensemble treatment resides in the fact that the multiplicity of the results also quantifies the spread of the model solutions, which provides useful information for the subsequent use of the model predictions for planning purposes or more generically decision-making as it is a measure of the variability of the options, scenarios or simply predictions.

When using ensembles in the realm of air quality modeling and atmospheric dispersion, the general tendency is to combine results of models that belong to the same category. Especially when referring to ensembles of opportunity (e.g. Galmarini et al. 2004; Tebaldi and Knutti al. (2007); Potempsky and Galmarini, 2009, Solazzo et al. 2012; Solazzo and Galmarini, 2015), which combine results from different models applied to the same case study, it is customary to consider as members those obtained from a homogeneous group of models. In particular, the scale at which models operate seems to be a discriminant in all such studies that have been performed to date. Therefore, meso-, regional-, and global-scale model results are grouped in ensembles according to their scale of pertinence. In air quality studies, this has been the case for example in Fiore et al. (2009), Solazzo et al. (2012), Kioutsioukis and Galmarini (2014), and Kioutsioukis et al (2016). Colette et al. (2012) analyzed as part of an analysis of the exposure in Europe, results from an ensemble of opportunity of a total of 6 models, 3 of which where global and 3 regional. The focus however was not the analysis of the contribution of neither the hybrid character of the group to the ensemble result nor the role of redundancy and reducibility of the set, but more obtaining a robust assessment of the 2030 air quality in Europe. A potential benefit of the mixed ensemble was spelled out there

but never verified in line with the opportunity character of the grouping. Therefore
there is no record in the literature of a study of an ensemble of models working at
different scales.

When developing a model, the scale selection is deeply rooted in the approach to
atmospheric modeling and it finds a theoretical justification in the alleged scale-
separation shown in the energy spectrum of dynamic variables such as horizontal or
vertical wind velocities (Van der Hoven, 1957). Although it is now well accepted that
the assumed scale separation does not have general validity, (e.g. Galmarini et al.
1999, Pielke, 2013) and especially not for scalars (e.g. Galmarini et al., 2000;
Michelutti et al., 1999; Jonker et al., 1999; Jonker et al., 2004), it has become a
convenient theoretical justification for the development of numerical models at
specific scales and to address the challenge that the computational solution of the
fundamental equation is imposing. Numerical constraints, in fact, oblige us to
identify the portion of the energy spectrum to be explicitly resolved by the model.
Larger domains imply larger grid spacing for practical constraints on the number of
grid points where the equations are to be solved. Larger domains on the one hand
allow us to move the resolved scales up in the atmospheric spectrum but at the
same time the coarser resolution leads to the loss of detail in the treatment of sub-
grid processes which are represented by parameterizations. Thus, for example, a
model that has the entire globe as simulation domain will have to use a horizontal
grid spacing of 25 to 100 km and therefore approximate (parameterize) the large
number of important processes occurring below those grid sizes. Conversely and
under normal conditions, a regional scale model that works with a horizontal grid
spacing of approximately 12-15 km will resolve explicitly the dynamics and transport
that occurs at scales larger than that distance but will not be able to extend the
computational domain to the hemispheric or the global scale. The scale separation
hypothesis states that the energy peak of boundary layer processes is isolated from
the rest of the spectrum, thus justifying their parameterization in a global model.
The same principle holds for a regional scale model. However, in the case of a
regional scale model, all the processes with scales falling in between 12-15 km and a
global-scale model grid-spacing (25-100 km) are resolved explicitly.

Although models are developed according to specific scales, nothing prevents us from combining them in an across-scale ensemble. What may appear to be just another attempt to combine model results for the sake of further and diversely populating an ensemble, has in fact a more rigorous motivation. Models working at different scales represent with different degrees of accuracy and precision different portions of the atmospheric spectrum and therefore processes. Our working hypothesis is therefore that by combining global and regional scale models into an ensemble, there is a high probability that they would complement each other across scales and consequently provide an improved ensemble performance compared to single scale ensembles.

Since in this study we are dealing with chemical transport models (CTM) we should also consider that chemical mechanisms span across a wide range of time scales. This could also constitute an element of diversity for these two groups of the models although the time resolutions for regional and global scale models are comparable. One could argue that in regional domains in particular, regional models essentially represent in detail the chemistry over a timescale of 10-days which then gets advected out and "reset". For example, differing representations of organic nitrate lifetimes and how long they sequester $NO_x$ in the system, impacts large scale $O_3$. Thus the difference in chemical mechanisms related to longer-lived species and multi-day chemistry could also introduce diversity and be another reason for exploring such an "across-scale ensemble".

Apparent ancillary elements that could also improve the ensemble results are for example the differences in emission inventories or in general sources of primary information, whose accuracy and precision cannot be guaranteed a priori or evaluated and that could contribute to the development of additional probable solutions.

As presented in the past, the diversity of modeling approaches is the element that favors a better ensemble product (Kioutsioukis and Galmarini, 2014; Kioutsioukis et

al., 2016). In this sense the combination of model results that focus on different scales and that account in a different form for the chemical mechanism has the potential to increase the value of an ensemble to which we will refer from now on as the *hybrid ensemble*.

The focus in this paper will therefore be on the analysis of the behavior of a hybrid ensemble. The variable considered is the ozone concentration measured and modeled for the year 2010 over the European continent. The analysis takes advantage of the unique opportunity offered by the HTAP2/AQMEII3 activity which brought together global and regional scale models to work on the same case study with a high level of coordination (Galmarini et al., 2017) as far as the input data are concerned.

In section 2, the observations and model results used in the analysis are presented in detail. In Section 3 the model results are characterized in the phase space to clearly establish whether the two scale groups do indeed account for different portions of the energy spectrum in a distinctly different way. Prior to analyzing the performance of the different ensembles, in Section 4 we evaluate the individual models against the measurements using conventional statistics as well as the newly developed error apportionment analysis presented by Solazzo and Galmarini (2016). Section 5 and 6 are dedicated to the analysis of the individual scale ensembles and the hybrid ensemble. Section 7 is dedicated to the comparison hybrid ensemble and single scale ensemble performance. The conclusions are discussed in section 8.

**2. The models used and the case study**

The set of models results considered and analyzed in this work are those that contributed to the HTAP2 and AQMEII3 modeling initiatives described in Galmarini et al. (2017).

HTAP2 is the second phase of the modeling activities of the Task Force on Hemispheric Transport of Air Pollutants (TF-HTAP) during which a community of

global scale CTMs performed a large number of simulations with the primary goal of investigating the transcontinental exchange of atmospheric pollutants (Dentener et al, 2010; Fiore et al. 2009). AQMEII3 is the third phase of the Air Quality Model Evaluation International Initiative (AQMEII, Rao et al. 2011) which brings together a community of European (EU) and North American (NA) regional scale modelers to work on coordinated case studies over EU and NA. For this third phase, the regional scale air quality modeling activity has been performed within HTAP2 framework. The coordination between HTAP2 and AQMEII3, as detailed in Galmarini et al. (2017), relates to the use of HTAP2 global model results as boundary conditions to the regional scale models and the use of the same anthropogenic emission inventory (Janssens-Maenhout et al., 2015) by both communities. The list of regional and global scale models analyzed in this work is presented in Tables 1 and 2 respectively. The simulations are for the year 2010 and the regional scale models were all initiated and received boundary conditions from the same global chemistry transport model C-IFS (Flemming et.al, 2015). C-IFS is also one of the global models that are part of the global model ensemble. Different meteorological drivers are used by the models as presented in the table thus adding an additional level of diversity to the groups, which is beneficial for any ensemble treatment. The two sets of models have been extensively evaluated (Solazzo et al. 2017; Solazzo and Galmarini, 2016; Jonson et al., 2018; Galmarini et al. 2018).

The analysis presented here focuses exclusively on ozone over the EU continent for which the largest abundance of models for the two groups is available and for which case we can take advantage from the fact that the models' performance has been analyzed with respect to other species elsewhere (Im et al., 2017). In the figures and tables resulting from our analysis, we shall not identify the individual models used since our goal is the identification of possible advantages in using hybrid ensembles rather than evaluating individual model results.

Hourly modeled concentrations of ozone were extracted by the modeling groups at European routine and non-routine sampling locations presented in Figure 1S of the supplemental material. Details on the networks used can be found in Solazzo et al.

(2012), Im et al. (2015), and Solazzo et al. (2017). Surface data were provided by the
European Monitoring and Evaluation Programme (EMEP; http://www.emep.int/)
and the European Air Quality Database, AirBase
(http://acm.eionet.europa.eu/databases/airbase). For the purposes of comparing
the ensemble performance with observations, only rural stations with data
completeness greater than 75% for the entire year and elevation above ground
lower than 1000 m have been included in the analysis. The total number of valid
time series used is 405. Only rural stations have been selected as the capture more
background signal than local effects. Including urban and suburban stations in the
analysis would penalise global model, which won't be able to capture local effects on
ozone.

**3. Preliminary analysis of the two groups of models**

**3.1 Spectral analysis of the global and regional model time series of ozone**
**concentrations**
One year of one-hour resolution ozone data allows us to produce detailed spectra
from the two groups of models and the measured concentrations. In Figures 1, the
individual power spectra of ozone (plotted against the period in days for easier
interpretation) from global and regional models are compared with the spectrum of
the measured ozone. The time series of the rural monitoring stations have been
averaged prior to producing the spectra. In all subsequent results the measured time
series should be interpreted as ensemble averages of all available rural monitoring
stations.

Since ozone is a scalar quantity, its spectrum grows monotonically in log-log scale as
expected (e.g. Galmarini et al., 2000), showing a distinct peak around a period of 24
hours, corresponding to the daily boundary layer evolution and photochemical
production of ozone. This peak is captured well by the two groups of model. The
global set tends to slightly underestimate the energy associated with this period with
only a single model that overestimates it. The regional scale models are evenly
distributed around the spectrum of the measured time series. The two groups

behave remarkably similarly at scales smaller than the daily peak. The majority of the models overestimate the energy but capture the slope of the measured spectrum. As expected, the spectra of the global models are more scattered but yet very well behaved. A weak second peak is visible between 30 and 50 days, which could be easily attributed to the synoptic variability. Solazzo and Galmarini (2016) demonstrated that it could indeed be connected to meteorology and/or removal by dry deposition. Moving up the period scale, after the daily peak, all regional scale model spectra are below the observed spectra a behavior that continues apart from a few exceptions up until the 60-70 day period range. Out of seven global models however, only 3 under- or over-estimate the energy in this period-range while the rest matches the observed spectrum. At 70-80 days a new peak appears in the observed time series, corresponding to the seasonal variability. Only one global model captures the observed time series, three models seem to anticipate it at smaller periods and even in the regional scale group there is a variety of behaviors including a monotonic increase of the energy throughout this period range.  Beyond the 100-day period the ozone energy spectrum grows monotonically, which the global model group matches the power line very closely whereas the regional scale group shows a more erratic behavior.

This first test is important to assess the fundamental differences between the two sets of models with respect to the characteristics of the signal, the periodicities present in the latter and the ability to reproduce the power or the variance of the measured signal at the various frequencies (periods).  In addition, it can give us an idea of the level of complementarity that exists between the two groups of models in the representation of the measured power spectrum. As clearly evident from Figure 1, both groups of models show an internal coherence in the representation of the power spectra. A remarkable result is the capacity of global models to represent the high frequency part of the ozone spectrum with an accuracy that is comparable with regional models. We can expect a complementarity in the behavior of the two groups in the large-scale energy range, which should be regulating the long-range transport and background values. The global models have a better representation of that portion of the spectrum than the regional one.

**3.2 Group performance and error apportionment**

A characterization of model performance for the individual members of the two groups beyond the information provided in Galmarini et al. (2018), Solazzo et al. (2017), and Jonson et al. (2018) is also appropriate at this stage.

The Taylor diagrams presented in Figures 2 provide an overview of the individual model performance across the year of reference. All model results underwent un-biasing (subtract the annual mean bias from the predicted hourly values, which produces a shift of the annual time series up or down by Mean Bias). We notice that the global models show a more scattered behaviour compared to the regional scale models, with performance distributed across a wider range of standard deviation values. Among the global scale models we find a clear outlier (model 5) whereas the rest tend to group in a rather narrow range of standard deviation values and correlations. Among the regional scale models we can also identify an outlier specifically model 9. The average RMSE values over all stations ranges from 22.4 to 25.9 ugm$^{-3}$ for the global models and 21 to 24.7 ugm$^{-3}$ for the regional models and are thus comparable. Global models overestimate the observed standard deviation while regional scale models with the exception of model 9 are evenly distributed across the observed values. The correlation coefficient is comparable for the two groups of models.

Figure 3 presents two classical skill scores for categorical events also applied by Kioutsioukis et al. (2016), namely the probability of detection (POD) and false alarm rate (FAR). The former represents the proportion of occurrences (e.g. events exceeding a threshold value) that were correctly identified, whereas the latter is the proportion of non-occurrences that were incorrectly identified as happening. In other words they measure *true* and *false positives*. In this case the scores are calculated on the basis of the individual model performances at each station. POD and FAR plots are presented as probabilities above breakdowns for different threshold values, where the abundance of the observed data per concentration

range is also given as histogram. A binned analysis of the RMSE demonstrates that
global models achieve lower RMSE at concentrations above 100ug/m$^3$; the opposite
is true for concentrations below this threshold. This partially explain the facts of
Figure 3.
At the same time the global models also have a higher percentage of false positives
as can be gleaned from the FAR index. This analysis is important to establish the
capacity of the models to simulate extreme values.

Using the methodology proposed by Galmarini et al. (2013), in Figure 4 we present
the decomposition of the model errors according to specific time scales. In this
figure, the individual model errors are shown as decomposed in the diurnal (<6h),
inter-diurnal (6h-1d), synoptic (1-10d), and long-term (>10d) time scales and the
residual. The decomposition is performed using a Kolmogorov-Zurbenko filter (Rao
and Zurbenko, 1997) applied to the Mean Squared Error (MSE) calculated from each
model and the observed ozone time series. Such analysis can be very revealing as it
identifies the scale and therefore the processes that are mainly responsible for the
deviation of the model results from the measurements as well as possible
persistence of errors at specific scales.

The figure reveals that most of the error is contained in the long term and diurnal
time scales. For regional-scale models, this is in agreement with the findings of
Solazzo and Galmarini (2016) and Solazzo et al. (2017). The same behaviour is also
found in the group of global models. What is remarkable is the similarity of the error
values at the diurnal time scale across the two groups. This suggests that the
difference in spatial resolution between the two sets of models does not seem to
influence the error at the scale at which atmospheric boundary layer dynamics and
daily emissions of ozone precursors are the dominant processes. Apart from few
exceptions (model 13 and 17 in the regional scale group and model 5 and 1 in the
global scale group), all other models have very comparable errors at that scale. A
comparable error across the two groups is found at the synoptic scale although this
is less surprising because this scale is explicitly resolved by the models in both groups
and strongly depends on the quality of the meteorological driver used. Since both
global and regional models employ assimilation of meteorological observations, they
are able to represent the synoptic scale comparably and are less dependent on
parameterizations employed. The long-term components have the largest error and
also show the most variability across models. Remarkably, the regional-scale models
seem to show smaller long-term error values than the global models although the
former show highly variable model-to-model errors. The strong dependence of the
long-term error on boundary conditions, (specifically lateral boundary conditions for
regional scale models and long range transport in the case of a global model, upper
air stratospheric intrusions and surface emission of ozone precursors and direct
ozone deposition) appears to influence the global scale group concentrations more
than the regional scale, though one should consider that almost all regional scale
models used boundary conditions from the same global model which nevertheless
does not have the smallest long-term error component of the error.

A useful pre-characterization of an ensemble can be obtained by the construction of
the Talagrand diagram (Talagrand et al. 1997). It is achieved by binning the range
from the minimum to the maximum modelled concentrations with as many bins as
the number of ensemble members plus one. The bins are then filled with observed
values based accordingly. For example, if an observed value is lower than the lowest
model value, it is assigned to the first bin, if it falls between the lowest and second-
lowest model value, it is assigned to the second bin, and so on. If it exceeds the
highest model value, it is assigned to the last bin. Figures 5 shows the Talagrand
diagrams for the global and regional and the regional+global set of models. The
figures reveal the tendency of the global model ensemble to be over-dispersed as
indicated by the accumulation of most of the observed data at the centre of
histogram and relatively few observations falling into the more extreme modelled
bins. The regional scale model ensemble shows a flat diagram which is an indication
of good group performance. A flat Talagrand diagram is an indication of the fact that
the group members equally cover (by proportion) all the observed range of values
and the group variability does not show an excess or deficiency in the number of
predictions in a specific range of observed values.

The first result obtained for a combined set of model results is shown in the third
panel of Figure 6, which presents the Talagrand diagram for the combination of the
two groups of models. Note that the number of bins (x-axis) has increased
corresponding to the new total number of models considered plus 1 (i.e. 7 global
models plus 13 regional models plus 1). The diagram for the combined group of
models qualitatively constitutes an improvement compared to those of the
individual group ensembles. The combination of the bell shaped diagram of the
global set with the relatively flat shape of the regional set produces a new
distribution within the range of modelled values of the observation  showing a flat
region between bins 5 and 18 and an under prediction region between bins 1 and 5
and 19 and 21, which now account for lower and higher values respectively
compared to the same bins of the global and regional sets.

**4. Analysis of the ensembles and building the hybrid one**

**4.1 Ensemble analysis per scale group**
Prior to analyzing the performance of the hybrid multi-model ensemble (mme_GR),
let us concentrate on the individual ensembles (mme_R and mme_G) of the two
groups for the sake of having an extra term of comparison beyond the measured
concentrations against which to compare mme_GR. In this study, we would also like
to build upon the research performed in other multi-model ensembles over the
years and rather than calculating only the classical model average or median
ensemble (mme) we shall also calculate three ensembles based on the findings from
Potempski and Galmarini (2009), Riccio et al. (2012), Solazzo et al. (2012); Solazzo et
al. (2013); Galmarini et al. (2013), and Kioutsioukis and Galmarini (2014). We shall
therefore refer to mmeS (Solazzo et al., 2012) as the ensemble made by the optimal
subset of models that produce the minimum RMSE; kzFO (Galmarini et al., 2013) as
the ensemble produced by filtering measurements and all model results using the
Kolmogorov-Zurbenko decomposition presented earlier and recombining the four
components that best compare with the observed components into a new model
set; and the optimally weighted combination mmeW (Potempski and Galmarini,
2009, Kioutsioukis and Galmarini, 2014, Kioutsioukis et al., 2016).

Figures 6 shows the effect of the various ensemble treatments for the two groups of
models separately and presented as Taylor diagram. The correlation has increased
and narrowed between 0.90 and 0.95 for both groups. As expected, the best
ensemble treatment of the two individual groups is mmeW which in the case of the
global models is comparable to mmeS and in the case of the regional scale models is
farther apart from mmeS. The fact that the optimal partition of the error in terms of
accuracy and diversity in an equal weighted sub-ensemble (mmeS) and the analytical
optimization of the error in a weighted full-ensemble (mmeW) are comparable for
the global models implies that this group better replicates the behavior of an
independent and identically distributed (i.i.d., represented by the square in all
pannels) ensemble around the true state set (on average). The range of
improvement of the RMSE is comparable for the two groups of models.

Of the entire set of ensemble treatments proposed, mmeS is the only one that works
with an identified subset of elements. The elements chosen in this context are those
that minimize a specific metric (e.g. RMSE). The combination of all possible
permutations of a pre-defined subset and for all possible subsets allows us to
identify the subgroup of models that performs best (Solazzo et al. 2012). This group
is the one that best reduces the redundancies and optimizes the complementarity of
the model results (Kioutsioukis and Galmarini, 2014). Other methods have been
devised to determine the optimal number of models (Bretherton et al., 1999; Riccio
et al. 2012) that are equally effective as the one used here, though they do not allow
identifying the members of the subset.  Beyond the use of the mmeS for the current
analysis, given the diversity in the number of models comprising the two ensembles
we have calculated the effective numbers of models (Bretherton et al., 1999) for the
regional and global sets in the attempt to verify whether the effective numbers were
close for the two sets. Figure 7 shows the $N_{eff}$ obtained for the global set and the
regional set. At over two third of the stations, the mmeS used 3-4 global models and
3-5 regional models. In other words, roughly half of the global models (3-4 out of 7)
produce the best result when constructing the mmeS globally while in the case of
the regional scale models less than half (3-5 out of 13) of all models are required.
Figure 7 also provides the frequency of contribution of the individual models to the
mmeS thus confirming the dominance of 3 global and 4 regional models determined
with the $N_{eff}$ analysis. What is presented in Figure 7 is the analysis for the aggregated
set of model results at all available monitoring points. We also would like to
determine the spatial variability of this result, i.e. to answer the question whether
$N_{eff}$ is uniform throughout the domain or whether there are sub regions that require
more or less models to construct mmeS.

In order to have a more objective assessment of the result presented in Figure 7 we
introduce a metric which samples only the diversity of the model results (see section
4.3). Following Pennel and Reichler (2011) and Solazzo et al. (2013) we introduce the
metric $d_m$ defined for *M* models at location *i* as:

$$d_{m,i} = e_{m,i}^* - R_{m,mme} mme_i^* \tag{1}$$

where
$$mme_i = \frac{1}{M} \sum_{m=1}^{M} e_{m,i} \tag{2}$$


$$e_{m,i} = \frac{mod_{m,i} - obs_i}{\sigma_{obs}} \tag{3}$$


and the * version of $e_{m,i}$ and $mme_i$ is obtained by normalizing them with $\sigma_e$ and
$\sigma_{mmei}$ respectively. $R_{m,mme}$ is the correlation between the individual and average
model results. Therefore only the uncorrelated portion of the individual result is
retained in d as measure of the diversity whereas the correlated portion is filtered
out. Applying this metric, the model results have been decomposed by means of the
Kolmogorov-Zurbenko filter described earlier and $N_{eff}$ has been calculated across the
domain for the most relevant components LT, SY, and DU. Figure 2S presents the
results for the two groups of models. For the long-term component, $N_{eff}$ results
shown in Figure 7 are largely confirmed with an overall spatial homogeneity of $N_{eff}$.
The global model set appears to require a larger number of models than the average
in critical areas like Northern Italy where the resolution would be insufficient to
capture the inhomogeneity of the concentration field due to the complex terrain in
that region (similarly in the western part of the domain). At the synoptic scale, the
regional scale models require slightly more models on average than the numbers
presented in Figure 7 and in some portions of the domain almost all available models
are required. The number of required models increases even further at the diurnal
scale. In the case of the global set, the average $N_{eff}$ is the same across these two
scales and more models are required in the Po valley (Italy) at the synoptic scale and
western Poland at the diurnal scale.


**4.2 Building the hybrid ensemble**
Given the fact that there is redundancy in the two groups of models and a disparity
exists in the overall and effective number of models in the two groups, a strategy has
to be devised so that no pre-determined weight is assigned to one of the two groups
thus masking the potential outcome of this study or creating false results. This goal is
accomplished by applying the following strategy.

We want to compare three equally populated ensembles of just global, just regional,
and mixed global and regional models. We will therefore reduce the ensemble of
regional-scale models and include extra models in the ensemble of global models
beyond the effective number calculated in Figures 7 and 2S so that the joint
ensemble will not be too small. In order to accomplish this, we select the global
models contributing most to the global ensemble beyond those identified by $N_{eff}$.
We begin by assuming that six is a reasonably abundant ensemble (as also indicated
by the effective number of regional scale models) and as such the single-scale
ensembles will be based on six members. Taking advantage of the various
techniques developed to build an ensemble presented earlier we define the
following sets:
- (mme_GR) hybrid ensemble of rank 6 (ensemble of 6 members) composed

of the best three global models and the best three regional models

- (mme_G) global ensemble of best six global models
- (mme_R) regional ensemble of best six regional models
- (mmeS_GR) optimally generated hybrid ensemble of rank 6 from the pool of

the best six global models and the best six regional models

- (mmeS_G) optimal global ensemble of rank 6
- (mmeS_R) optimal regional ensemble of rank 6
- (mmeW_GR) weighted hybrid ensemble composed from the best three

global models and the best three regional models

- (mmeW_G) weighted global ensemble of best six global models
- (mmeW_R) weighted regional ensemble of best six regional models

Among them, the mmeS_GR is the only ensemble product that allows unbalanced
contributions from global and regional models.

**4.3 Comparing the single scale multi-model ensembles with the hybrid one**
The comparison of the ensemble performances will be restricted to the months of
June -August when the photochemical production of ozone is at its maximum and
the number of exceedances is expected to peak throughout the continent. The
results of the comparison of the mme, mmeS and mmeW for the regional (_R),
global (_G) and hybrid cases (_GR) are shown in Figures 8. The elements common to
the three panels are:

• The hybrid ensemble of rank 6 composed of the three best global models and

the three best regional models (mme_GR) when compared to mme_G (best

six global models) and mme_R (best six regional models) does not show

improved performance, rather its skill is inferior to both mme_G and mme_R.

• For the other two kinds of ensemble treatments (mmeS and mmeW), the

combination of global and regional models produces some improvement

compared to just the global or regional ensembles in terms of correlation

coefficients, standard deviations and RMSE.


The partition of global and regional models in mmeS (Figure 9) shows that the
contribution of regional models is more frequent. Specifically, at two thirds of the
stations, the optimum hybrid ensemble of rank 6 consists of one or two global
models and five or four regional models, respectively. At only 15% of the stations,
mmeS consists of an equal number of global and regional models. The maximum
number of global models in the mmeS_GR ensemble is four, achieved at roughly 1%
of the stations. Conversely, at around 10% of the stations the hybrid ensemble
utilized only regional models.  The second panel of Figure 9 also gives the regional
distribution of the number of Global models contributing to the hybrid ensemble
clearly indicating a deficiency in the northeastern part of the domain.

In Figures 10,  POD and FAR show a net improvement over the mmeW_G results
when the hybrid ensemble is considered, with a minimum in false positives and a
maximum in true positives that closely match the mmeW_R results

The real improvement of the hybrid ensemble with respect to the single scale model
ensembles becomes evident when analyzing Figure 11. The pannels in the figure are
the collective representation of three of the most important characteristics of an
ensemble as proposed by Kioutsioukis and Galmarini (2014), i.e. diversity, accuracy
and error. On the x and y axes respectively "*diversity*" and "*accuracy*" are presented.
The former represents the average square deviation of the single models from the
mean of the models, whereas the latter is the square of the average deviation of the
individual model results from the observed value. As presented by Krogh and
Vedelsby (1995), the difference of the *diversity* and *accuracy* defines the quadratic
deviation of the ensemble average from the observed value. From the definition it
follows that in order for the ensemble result to be closer to the observed value one
has to find the right trade off between *accuracy* and *diversity* (A-D). A mere increase
in *diversity* does not guarantee a minimization of the ensemble error since it might
produce a reduction in the *accuracy*. What one hopes to obtain is the right
combination of models that provides the maximum *accuracy* and maximum
*diversity*. In the plots of Figure 11, the optimal condition is achieved when the model
results concentrate in the upper left quadrant of the plot toward the
(x=100/(Number of Models),y=1) point. In the plot, the accuracy parameter is
presented as deviation from the best model performance. The dots represent the
estimate of the two parameters at every location where measurements are
available. The colour scale is based on the RMSE. The two upper panels give the A-D
mapping for the mme_R and mme_G ensembles; the lower two panels give the map
for the hybrid ensembles, i.e. mme_GR + mmeS_GR. The difference in nature of the
two ensembles is clear form the two panels. Ensemble mme_G is more diverse and
accurate than mme_R (x values:69 in G and 66 for R, y: 0.75 in G, 0.66 in R). The
combination of the two produces a decrease in the two parameters (GLO+REG
(mme6)). However, if the models are selected as in mmeS, both accuracy and
diversity increase (GLO+REG (mmeS6)). The real advantage of the combination is
visible in a slight increase of the diversity as compared to mme_GR and a marked
improvement of the accuracy from 0.71 to 0.81. The error decreases from a median
value of 17.9 to 15.6 and from an Inter Quartile Range of 5.1 to 3.8.
In Figures 12 the spectra of the ensembles are presented. For the just global, just
regional-scale ensembles and the rank 6 hybrid ensemble, the spectra of mme,
mmeS, mmeW and kFO are shown in the figure.  Figure 12 also shows the spectra of
the four ensembles, mme_R6, mme_G6, mme_GR6 and mmeS_GR6 for which the
largest six contributors from the regional models, the six global, and three regional
plus three global models were used. From the picture we see that regardless of the
treatment, the ensemble data capture the ozone power spectrum with no notable
deviation from the measured spectrum. It is important to note that an ensemble
treatment is a purely statistical treatment that does not consider any physics
constraints. The deficiencies that were originally present in the individual model
spectra are still present in the ensemble results, particularly the large power deficit
in the range from 0.8 days to 100 days. The mme_GR spectrum appears to produce a
slight improvement toward filling this energy gap, but the change is very small.

**5. Discussion and conclusions. How much is the improvement attributable to the**
**hybrid character of the ensemble?**
The analysis presented above gives us clear indications that the combination of the
two sets of models analysed produces an improvement in the ensemble
performance. In particular, the hybrid ensemble appears to be superior to any
single-scale ensemble in the optimum setting. For example, given six global, six
regional and three global and three regional ensembles, the optimization always
favours the hybrid ensemble. This was repeated for all examined cases: the annual
hourly records, the JJA hourly records and the annual daily maximum records.
- The improvement is in the range 1-5% compared to single scale optimum
ensembles
- POD/FAR show a remarkable improvement, with a steep increase in the
largest POD values, though comparable to the other for the hybrid ensemble
and comparatively smallest values of FAR across the concentration ranges.

Some important considerations need to be made at this point. It is difficult to find
quantitative evidence for the fact that the hybrid ensemble improvement can be
unequivocally attributed to the multi-scale nature of the ensemble. We have no
evidence, nor guarantee, that the same kind of improvement could be reached by
adding more regional-scale models to the regional-scale ensemble, or more global
models to the global-scale ensemble. However, what is a clear conclusion is that the
regional-scale ensemble is characterised by a higher level of redundancy in the
members than the global ensemble, since less than half of the members produced
the optimal ensemble, and that the use of the three best members from the
regional-scale ensemble and three best global-scale models produces an
improvement in the ensemble performance. This last argument suggests that the
addition of more model results of the same "nature" would just contribute to further
increase the level of redundancy, while on the other hand, the improvement
obtained could indeed be attributed to the different "nature" of the global-scale
models compared to the regional-scale models.

Therefore, considering:
• the large number of regional scale models and the spectrum of diversity in
their nature (only a small number of the same models were used by multiple
groups and there was an abundance of models developed independently
from one another);
• the relatively smaller number of global model results compared to the
regional models and also their different nature;
• the fact that the two groups of models used the same emission inventories
and all the regional scale models used boundary conditions from the same
global model;
one could attribute the improvement of the mmeS_$GR$ ensemble performance to
the difference in nature of the two groups and a complementary contribution of the
two toward an improved result.


**Acknowledgments**
The group from University of L'Aquila kindly thanks the EuroMediterranean Centre
on Climate Change (CMCC) for the computational resources. P.T. is beneficiary of an
AXA Research Fund postdoctoral grant. We acknowledge the EC FP7 financial
support for the TRANSPHORM project (grant agreement 243406). CIEMAT has been
financed by the Spanish Ministry of Agriculture and Fishing, Food and Environment.
DKH and YD recognize support from NASA HAQAST. The UMU group acknowledges
the Project REPAIR-CGL2014-59677-R of Spanish Ministry of the Economy and
Competitiveness and the FEDER European program for support to conduct this
research. The views expressed in this article are those of the authors and do not
necessarily represent the views or policies of the U.S. Environmental Protection
Agency. The MetNo work has been partially funded by EMEP under UNECE.
Computer time for EMEP model runs was supported by the Research Council of
Norway through the NOTUR project EMEP (NN2890K) for CPU, and NorStore project
European Monitoring and Evaluation Programme (NS9005K) for storage of data. RSE
contribution to this work has been financed by the research fund for the Italian
Electrical System under the contract agreement between RSE S.p.A. and the Ministry
of Economic Development – General Directorate for Nuclear Energy, Renewable
Energy and Energy Efficiency in compliance with the decree of 8 March 2006.

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

**Figure Captions**

Figure 1S: Spatial distribution of the 405 rural monitoring stations where ozone
model results where produced and observations were available

Figure 1: Power spectrum of observed ozone (thick line) obtained from the average
one year time series across all measuring locations and of global models and regional
models.

Figure 2: Taylor diagram of Global models and regional models

Figure 3: Cumulated Probability of detection (POD) and False alarm rate (FAR) for
Global and regional models at various ozone concentration threshold

Figure 4: Distribution of the Mean Square Error (MSE)  across the models of the two
communities and the scales in which the signal has been decomposed (LT, long term;
SY synoptic; DU diurnal; ID inter diurnal; see text for definition)

Figure 5: Talagrand diagrams of Global models, Regional models and the Global +
Regional set of model results

Figure 6: Taylor diagram of the four ensemble treatments considered in the text
obtained from the global and regional models

Figure 7: Effective number ($N_{eff}$) of models calculated according to Bretherton et al.
(1999) for the two groups of models; and frequency of contribution of each model to
the mmeS

Figure 2S: Number of effective models for the two groups obtained at all monitoring
locations considered thus giving the spatial structure of the ensemble size and for
three of the four components in which the modelled time series have been
decomposed, namely: LT, SY and DU.

Figure 8: Comparison of the performance of three ensemble treatments (mme,
mmeS and mmeW) for three groupings of models (regional *R*, global _G, and mixed
global and regional _GR)

Figure 9: Contribution of Global models to mmeS_GR and its spatial representation

Figure 10: POD and FAR for the best performing ensemble treatment (mmeW) and
for three ensemble grouping (regional *R*, global _G, and mixed global and regional
_RG)

Figure 11: Representation of the accuracy (y-axis) vs diversity (x-axis) and RMSE for
the ensemble of the most present 6 global and regional models respectively (top
row) and a hybrid ensemble calculated with mme and mmeS ensemble methods
(bottom row). For reference, the square represents the ideal point corresponding to
an independent and identically distributed models (i.i.d ensemble). If the models are
i.i.d. then all eigenvalues are equal, each explains 1/N of the variance and therefore
for 6 models the point is at (0.16; 1).


Figure 12: Spectra behaviour of the ensemble treatments: full global ensemble (top);
full regional ensemble (middle); mme of 6 most frequently present global and
regional models and the hybrid ensemble calculated with mme and mmeS ensemble
methods (bottom)
















**TABLE 1. PARTICIPATING REGIONAL MODELLING SYSTEMS AND KEY FEATURES. THE DARK SHADED CELLS CONTAIN INFORMATION ON MODELS THAT WORKED OVER THE NA DOMAIN THEOTHERS ON THE EU ONE**


| Operated by | Modelling system | Horizontal grid | Vertical grid | Global meteo data provider | Gaseous chemistry module |
|---|---|---|---|---|---|
| Finnish Meteorological Institute (working with 2 versions) | ECMWF-SILAM_H, SILAM_M | 0.25 x 0.25 deg (LatxLon) | 12 uneven layers up to 13km. First layer ~30m | ECMWF (nudging within the PBL) | CBM-IV |
| Netherlands Organization for Applied Scientific Research | ECMWF-L.-EUROS | 0.5 x 0.25 deg (latxlon) | Surface layer (~25m depth), mixing layer, 2 reservoir layers up to 3.5km. | Direct interpolation from ECMWF | CBM-IV |
| University of L'Aquila | WRF-WRF/Chem1 | 23 km | 33 levels up to 50hPa. 12 layers below 1km. First layer ~12m | ECMWF (nudging above the PBL) | RACM-ESRL |
| University of Murcia | WRF-WRF/Chem2 | 23 x 23 km$^2$ | 33 levels, from ~24m to 50hPa | ECMWF (nudging above the PBL) | RADM2 |
| Ricerca Sistema Energetico | WRF-CAMx | 23 x 23 km$^2$ | 14 layers up to 8km. First layer ~25m. | ECMWF (nudging within the PBL) | CB05 |
| University of Aarhus | WRF-DEHM | 50 x 50 km$^2$ | 29 layers up to 100hPa | ECMWF (no nudging within the PBL) | Brandt et al. (2012) |
| Istanbul Technical University | WRF-CMAQ1 | 30 x 30 km$^2$ | 24 layers up to 10hPa | NCEP (nudging within PBL) | CB05 |
| Kings College | WRF-CMAQ4 | 15 x 15 km$^2$ | 23 layers up to 100hPa, 7 layer below 1km. First layer ~14m | NCEP (Nudging within the PBL) | CB05 |
| Ricardo E&E | WRF-CMAQ2 | 30 x 30 km$^2$ | 23 VL up to 100hPa, 7 layers < 1km. 1st @~15m | NCEP (nudging above the PBL) | CB05-TUCL |
| Helmholtz-Zentrum Geesthacht | CCLM-CMAQ | 24 x 24 km$^2$ | 30 VL from ~40m to 50hPa | NCEP (spectral nudging above f. troposhere) | CB05-TUCL |
| University of Hertfordshire | WRF-CMAQ3 | 18 x 18 km$^2$ | 35 VL from ~20m to ~16km | ECMWF (nudging above PBL) | CB05-TUCL |
| INERIS/CIEMAT | ECMWF-Chimere_H Chimere_M | 0.25 x 0.25 deg | 9 VL up to 500hPa. 1st L @~20m | Direct interpolation from ECMWF | MELCHIOR2 |












**TABLE 2. PARTICIPATING GLOBAL MODELLING SYSTEMS AND KEY FEATURES.**

| Operated by | Modelling system | Horizontal grid (km x km or °lat x° lon) | Vertical grid | Global meteo data provider | Gaseous chemistry module | References |
|---|---|---|---|---|---|---|
| NAGOYA, JAMSTEC, NIES | CHASER_re1 | 2.8°x2.8° | 32 VL up to 40 km | ECMWF (nudging above PBL) | Sudo et al. (2002) | Sudo et al. (2002), Watanabe et al. (2011) |
| NAGOYA, JAMSTEC, NIES | CHASER_t106 | 1.1°x1.1° | 32 VL up to 40 km | ECMWF (nudging above PBL) | Sudo et al. (2002) | Sudo et al. (2002), Watanabe et al. (2011) |
| ECMWF | C-IFS | Ca. 80 km | 60 VL from surface to 0.1 hPa – lowest level 15 m | IFS | CB05 | Flemming et al. 2015 |
| MetNo | EMEP_rv4.8 | 0.5° x 0.5° | 20 uneven layers up to 100hpa. First layer ~90m | ECMWF IFS dedicated model run | EMEP | Simpson et al. 2012 http://emep.int/mscw/mscw_publications.html |
| Univ. Tennesee | H-CMAQ | 108 km x 108 km | 44 layers up to 50hPa | WRF | CB05 | Xing et al. (2015) |
| Univ.Col. Boulder | GEOSCHEM-ADJOINT | 2° x 2.5° | 47 levels up to 0.066 hPa (bottom of the last grid) | GEOS-5 | GEOS-Chem | Henze et al. (2007) |
| US-EPA | H.-CMAQ* | 108kmx 108km | 44 lev to 50hPa | WRF nuged with NCEP/NCAR | CB05TUCL | Mathur et al. (2017) |

* H-CMAQ is strictly a hemispheric model but for the purposes of this analysis is expected to
behave the same as global models over the EU domain, therefore, for the rest of the paper
we will refer to it as "global models".

Figure1

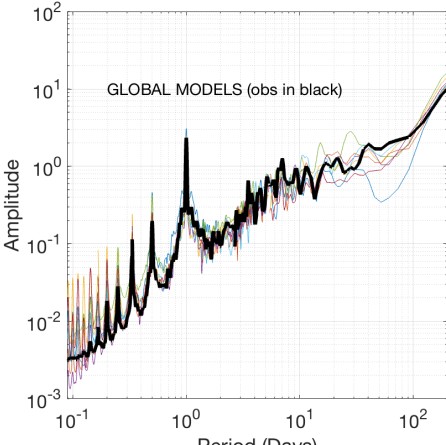
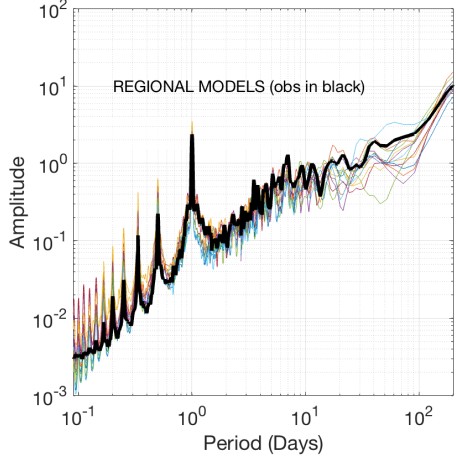

Figure 2

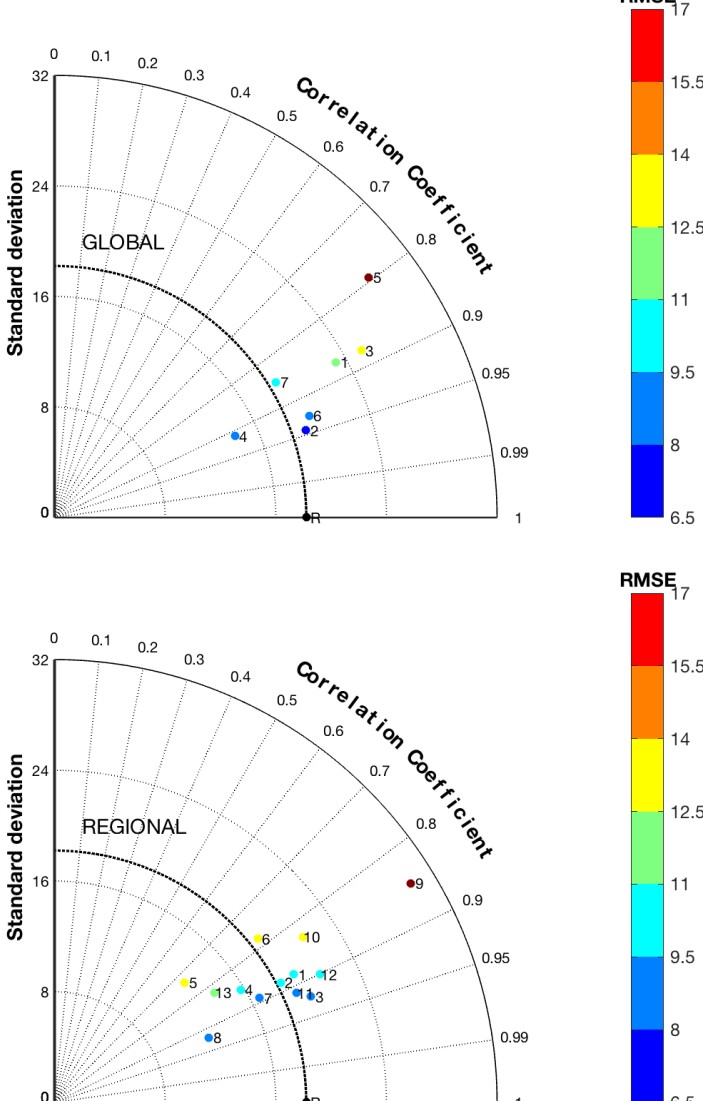

Figure 3

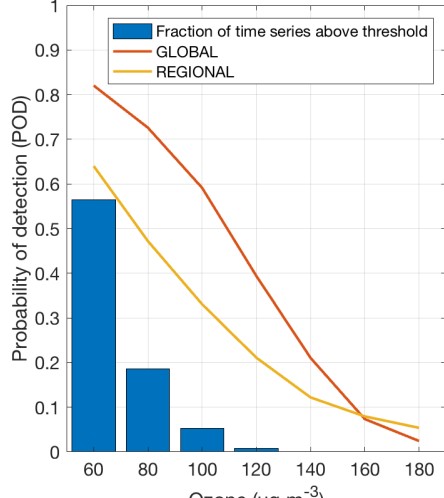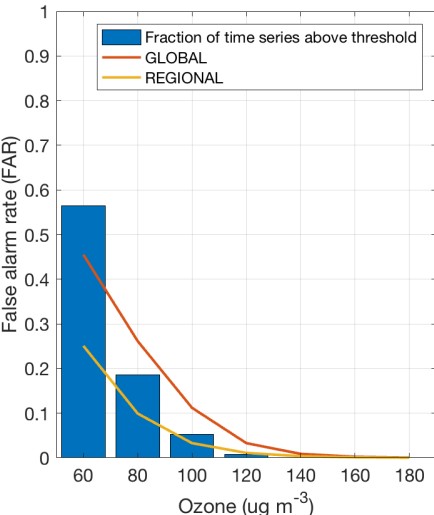

Figure 4

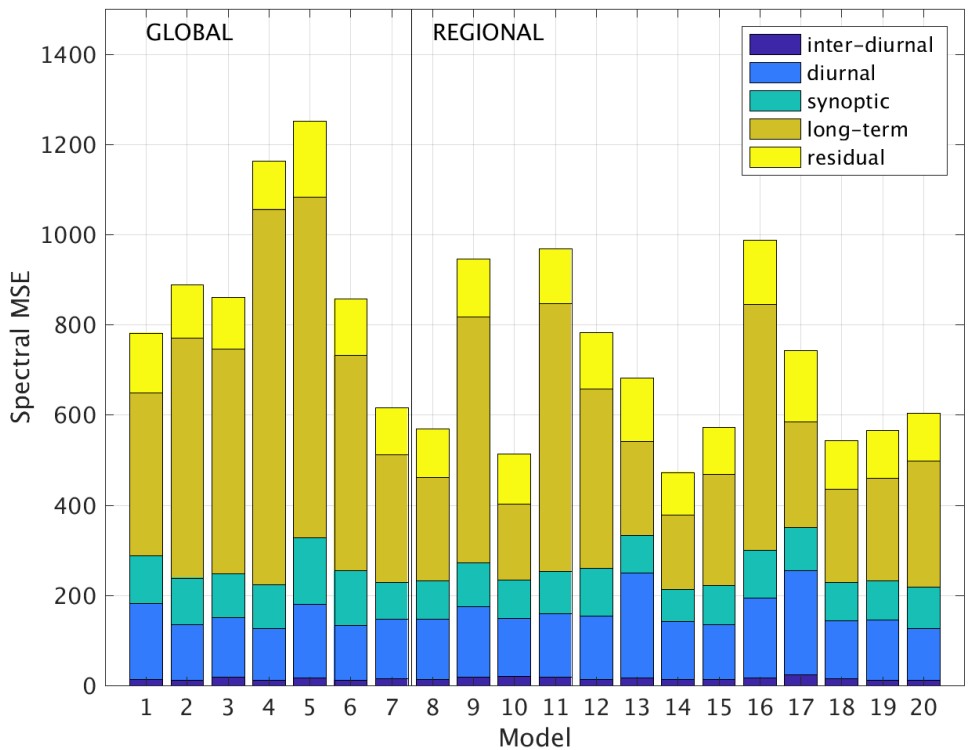

Figure 5

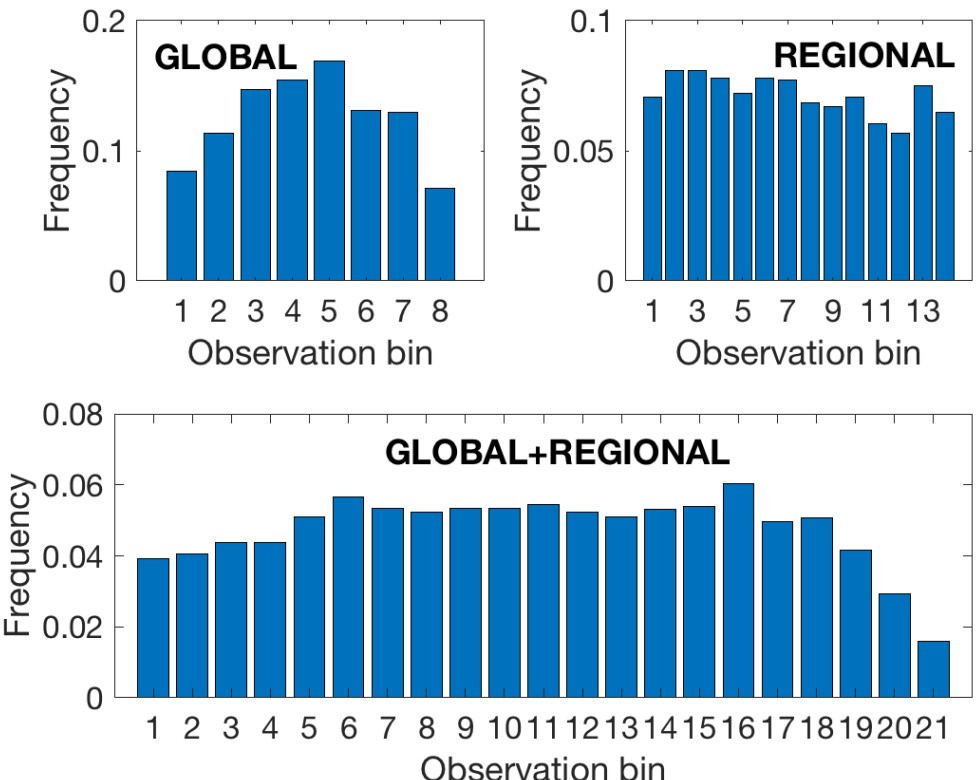

Figure 6

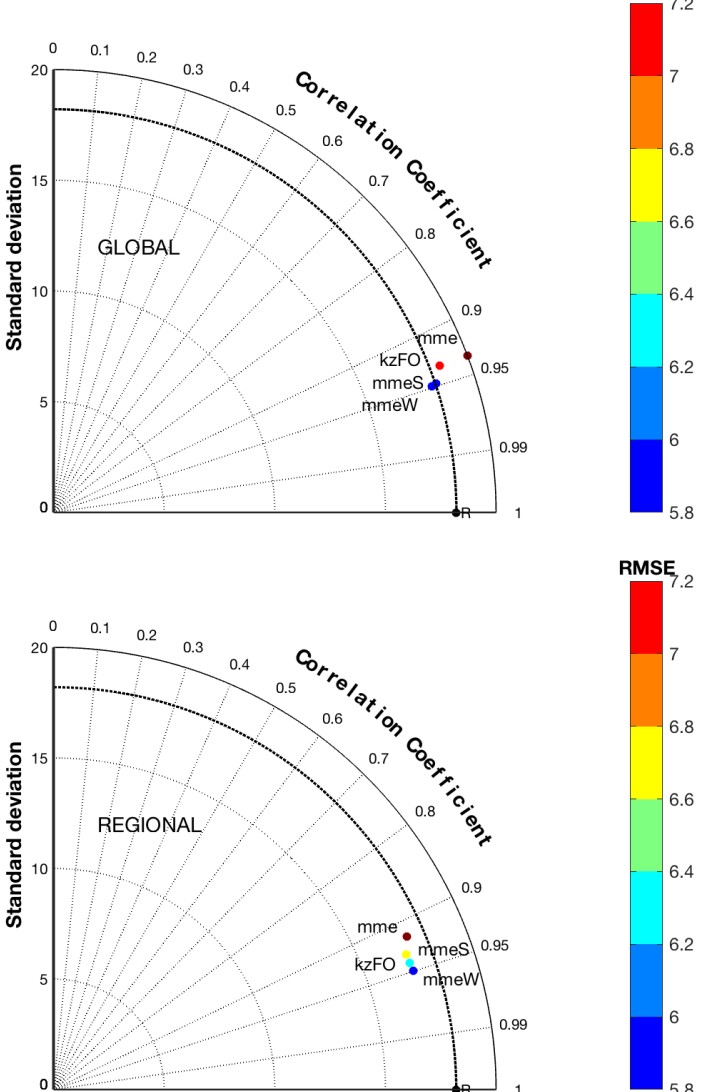

Figure 7

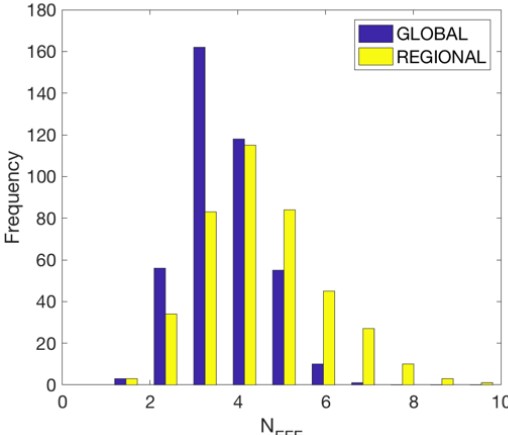
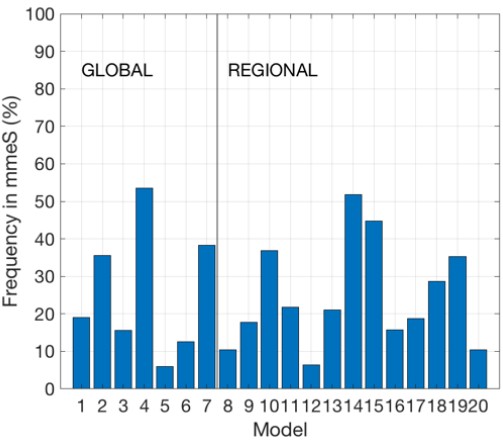

Figure 8

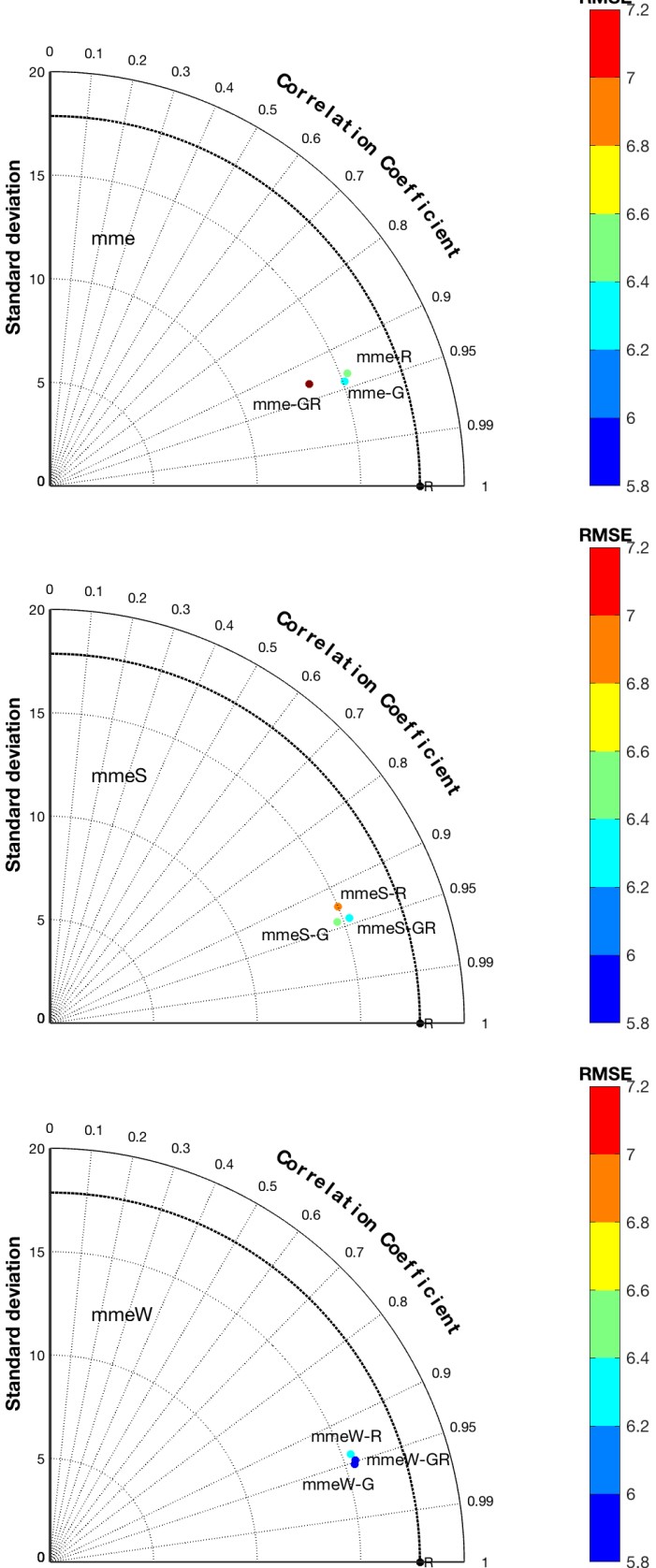

Figure 9

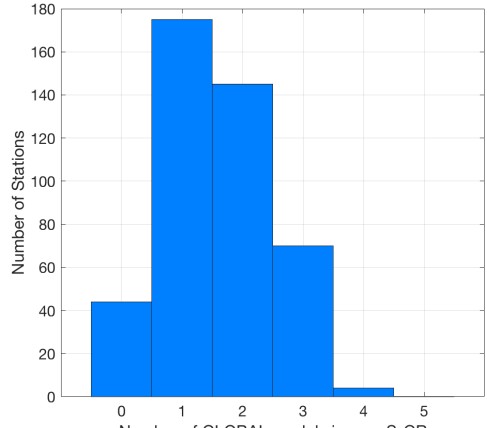 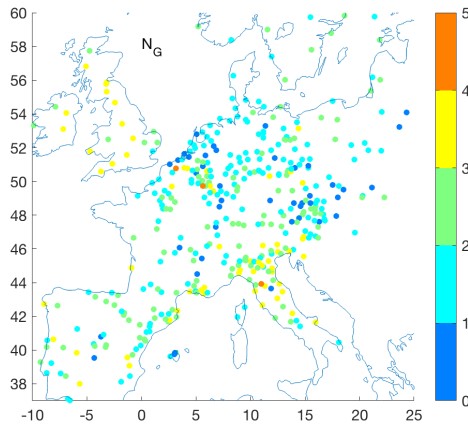

Figure 10

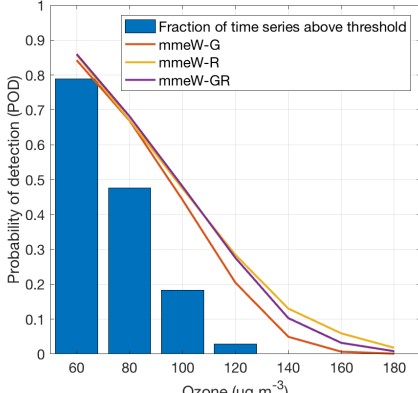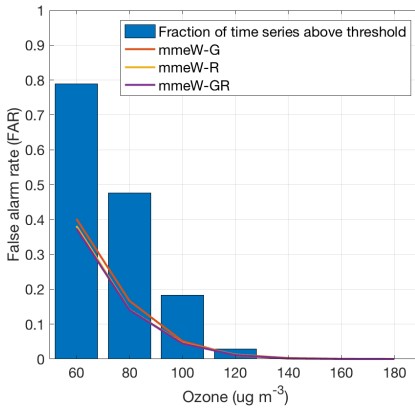

Figure 11

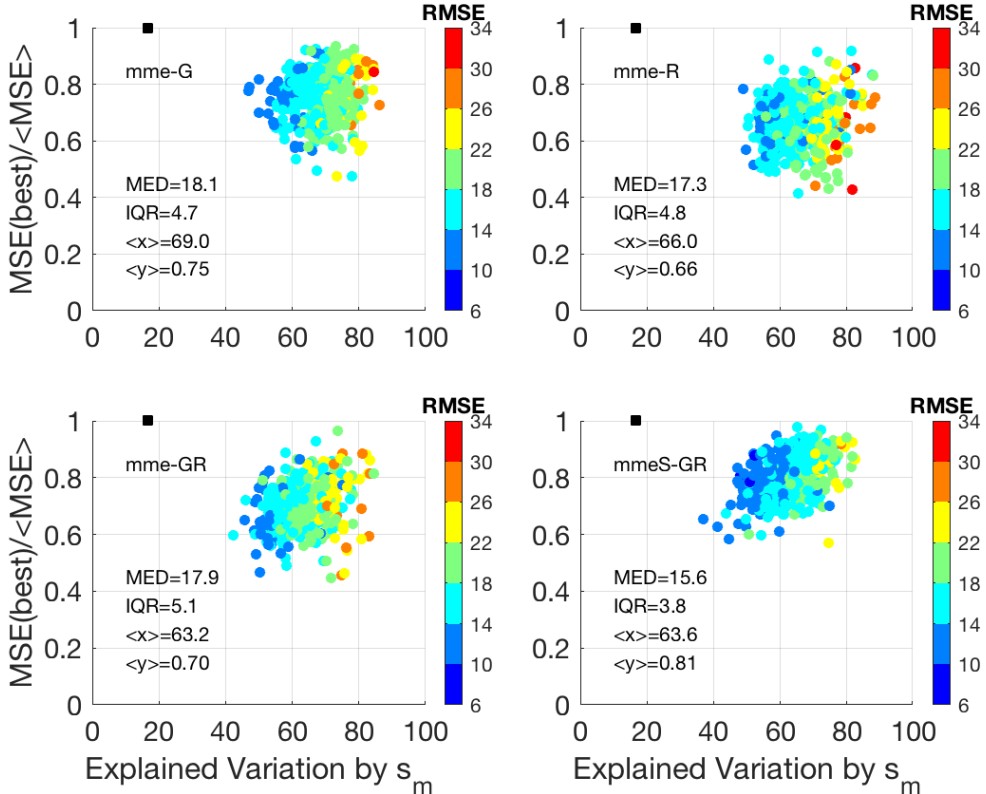

Figure 12

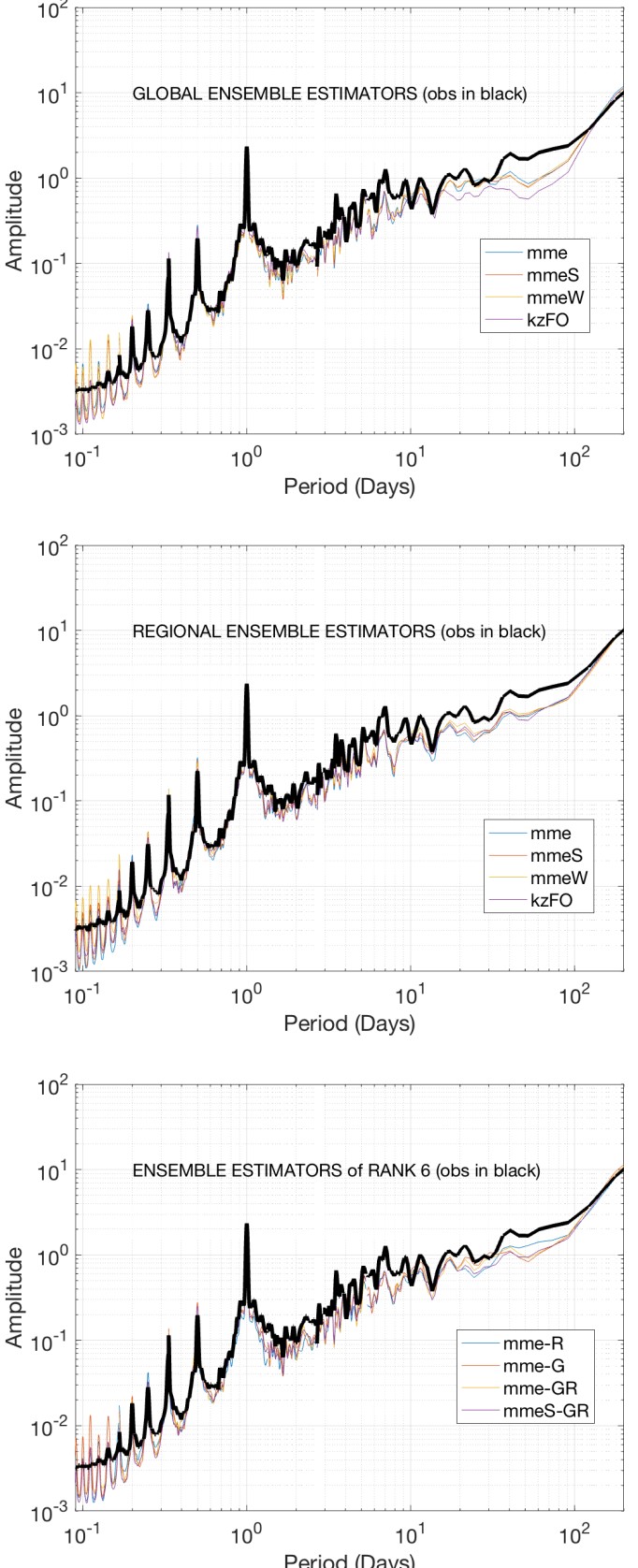