# Peer review of "Two-scale multi-model ensemble: Is a hybrid ensemble of opportunity telling us more?"

_Atmospheric Chemistry and Physics, 2018_

## Referee Comment (RC1) · Anonymous Referee #2 · 5 Mar 2018

Galmarini et al. show a statistical analysis of the performance of a multi-model ensemble of regional and global-scale chemistry-transport models in reproducing ozone surface concentrations over Europe. The simulations have been conducted as part of phase 3 of the AQMEII-HTAP initiative. A number of methods are introduced to construct optimal sub-ensembles for only global, only regional, and hybrid ensembles, and the performance of these is pitted against each other. In general, they show a small improvement of the optimally-selected hybrid ensemble over the pure global and pure regional ensembles. A cautious conclusion is drawn that the multi-scale nature of the hybrid ensemble actually proves to be beneficial.

General comments:

My general suggestion to the editor is to publish this manuscript after a number of

points have been addressed, as the analysis seems to be sound, the methods are reasonably well explained, most of the discussion is reasonable, and the conclusions drawn are sufficiently cautious. I have given it "major revisions" in the hope to see it again before it is published, but in general the manuscript is very close to "minor revisions" for me.

It should be noted that surface ozone concentrations (and their observations) are a notoriously difficult measure to analyse and compare against models due to the strong effects of local emissions and deposition, so care should be taken to limit the scope of this manuscript to the performance of the ensemble in terms of surface ozone. This should be discussed in the introduction.

Before going into details I would finally urge the authors to reconsider putting 14 (!) figures into this manuscript and rather move several of them to a supplement in order to improve readability.

Specific comments:

P2, L70 define which "spectra" you refer to here.

P2, L74-75 is it a good thing that you use reg-glob equally at only 15 % of stations?

P5, L60-66 this is a slightly confusing amalgamation of arguments. Limitations in space are combined with different representations of gas-phase chemistry. I suggest rephrasing this paragraph and simply discussing gas-phase chemistry mechanism diversity.

P7, L221: regional models biased towards C-IFS?

P7, L229: "can take stock" seems inappropriately used in this context, revisit expression.

Figure 2: please use the same axis (Period in days is fine) for 2a as for 2b and c.

P8, L249ff: why did you average the spectrum? how did you average the spectrum? Are 24h peaks as pronounced in the model as in the measurements?

P8, L249ff: Spectrum analysis also results in mangling daily maxima and daily minima. I am not sure if I like the fact that this means you are evaluating both photochemistry (daily maxima) and representation of boundary layer parameterization (nighttime titration)... Could be worth discussing in the text.

P8, L255: "The time series of the rural monitoring ...", these are simply your "observations", right? Not a subset or something? I suggest replacing "rural monitoring stations" with observations so as not to confuse readers.

P8, L268: Interesting! Do you dare to speculate as to why this is?

P8, L269-271: "A weak sec..." this sentence seems repetitive of what comes afterwards, remove or merge with remainder of paragraph.

P9, L298-300: "An element of surprise...". Surprising sentence - it reads like the introduction to the paragraph before. Why is it at the end? Is it a conclusion? Might want to rephrase.

P11, L337-338: if it is important, please tell us what you think about GLO vs REG now. Is the fact that GLO have a higher POD but also a higher FAR a good thing? Not really, no?

P12, L371: "... transport in the case of a global model, " are you not talking about regional models here? Consider cleaning up the paragraph.

P13, L401ff: I disagree with this assessment of the combined histogram (Talagrand diagram). The regional models (6b) actually came quite close to the ideal Talagrand diagram (equal distribution). Combining them and increasing bin number does not (necessarily) increase the value range. Now you still have an overdispersed model system, which just happens to be more correctly distributed amongst values closer to the mean (bins 5-18). I suggest to rephrase this paragraph.

Fig 9: very small panels, please improve this to make it readable.

P17, L532: Figure 11 does not have panels a, b and c...

P17, L539ff: calling this improvement "systematic" is an exaggeration - there is no estimate of the uncertainty of these numbers, hence you have no idea whether the difference ("improvement") is statistically significant. This should be rephrased and written more cautious.

Fig 13, and also P18, L572-573: it would be very helpful if the optimal point would be marked in these plots. If I understand this plot correctly, for a 6 member ensemble the optimal point is (x=16,667, y=1; description in text says in line 573: (x=100/(Number of Models),y=1) with number of models = 6). All points are pretty far off of this optimal point in all the plots!

P18, L579: typo "form"

P18, L580ff: y scales 0-1, hence the "y values" cited here are for x, probably.

P18, L583: I guess you are talking about Figure 13d (mmeS6) as being superior here - rephrase to make this clear.

Fig 14: legends miss "obs" description (if that is what the thick red line is supposed to show)

P19, 587-600: this paragraph deserves its own (sub)caption, as it concerns a quite different and important point than the previous text

---

## Referee Comment (RC2) · Anonymous Referee #1 · 13 Mar 2018

The researchers aim to evaluate whether a hybrid model ensemble, using a combination of both regional and global models, can better reproduce observed variations in surface ozone than an ensemble made up of only global or only regional models. In order to do so, they take advantage of existing, global-scale model output produced during the second Hemispheric Transport of Air Pollution modeling experiment (HTAP2), while using regional model output produced from the third phase of the Air Quality Model Evaluation International Initiative (AQMEII3). Model output is compared to a full year of hourly ozone data from multiple monitoring stations across Europe. Applying a variety of interesting and appropriate analysis techniques, the authors find that the use of a hybrid rather than single-scale ensemble can yield an improvement in performance on all three metrics.

[Figure]

The question being addressed is both interesting and important, and the methods used are appropriate. The magnitude of the advance is limited, in that this is a methodological advance which has specific relevance only to ensemble modelers, but in that field presents a significant finding. However, the paper is significantly hampered by its presentation. Although the content of the paper seems generally of high quality, the problem of presentation is sufficient that I recommend major revisions be made before the paper be considered for publication. I have given major issues below in paragraph form, followed by an itemized list of minor issues.

The first issue is the structure of the paper, in particular the use of figures. Although the manuscript clearly has a narrative, it is lost in the volume of analysis: 14 figures, usually with multiple sub-panels, and presented without any subsections to help structure the paper. I would urge the authors to move a significant fraction of both the figures and the analysis into the supplemental information, and retain only the most interesting and relevant findings in the main text. The paper would also benefit from prudent use of headings and sub-headings. Sections 3-4 should really be subsections 3.1 and 3.2, discussing the individual model performance without considering ensembles. Sections 5-7 then cover performance at the ensemble level as sections 4.1 through 4.3, with section 8 becoming the paper's conclusions. Even if the authors do not take up this suggestion, it would help the readability of the manuscript if they included a brief discussion at the end of what is currently section 4 to summarize the ways in which global models are generally providing better or worse results compared to regional models.

The second issue is the presentation of the figures. I would usually consider this to be only a minor issue, but in this case the figures are so minimally labeled or polished that it compromises the readability, and therefore the quality, of the manuscript of a whole. Figure 2 provides a good example. First, the upper panel undergoes no serious analysis, and the information presented in it is redundant as it is presented again with smoothing in the next two panels. Second, the data in the remaining two panels have

no legend, making it difficult to distinguish at first glance between the model results and those from the observations. Third, the vertical axis is labeled only as "|Y(f)|", which is both obscure and incomplete, lacking any units. Beyond just incomplete or redundant information, there are also several stylistic choices which make the figures difficult to interpret. The panels are labeled only as GLO and REG; it seems like it would have taken minimal effort to label these more naturally as "Global" and "Regional". I would also suggest using more natural tick choices on the x-axis (e.g. 1 hour, 12 hours, 1 day, 1 week, 30 days, 90 days, 1 year) and removing the clutter of the grid lines.

Each of the other figures has similarly strange choices. Figure 3 shows tick marks on the color bars which do not correspond to the color bands, the limits on the standard deviation change between sub panels, and the sub panels are completely unlabeled. Figure 4 has ample space for a full vertical label (i.e. "Probability of detection (POD)" and "False alarm rate (FAR)") but instead uses the acronyms POD and FAR, while still using the terse bar labels GLO and REG. Figure 4 also has two lower sub-panels with much smaller fonts than the upper panels, an X-label "Ozone" which lacks either units or temporal period, and a bar series with the name "##%". Similarly, figure 5 uses the bar labels ID, DU, SY, LT, and un; although four of these are explained in the caption, there is plenty of space in the figure itself to include the full labels ("inter-diurnal", "diurnal", "synoptic", "long-term", "residual"), which would make the abbreviations redundant as they are never used in the text. Figure 6 shows Talagrand diagrams, which are difficult to interpret for unfamiliar readers and deserve special care. Unfortunately, as presented the figure has no obvious meaning, with 3 unlabeled panels and vertical axes labeled only as "Frequency", while the X-axis label "Model id" is misleading. "Model id" implies parity with the "Model id" label in other figures, whereas it really refers to "(n-1)th to nth lowest model ozone concentration". Similar criticisms are applicable for the remaining figures.

The third issue is that the abstract lacks any quantitative conclusions regarding the paper. I recommend that the authors consider rewriting the abstract to include more of the

information provided in the paper's conclusions, with a particular focus on quantitative outcomes (e.g. the 1-5% improvement observed relative to single-scale ensembles when considering a specific metric).

The remaining issues are relatively minor, and are listed below:

- In the conclusions the authors refer to analysis of annual hourly, JJA hourly, and annual daily maximum records. However, until that point there seems to be no discussion of the latter two metrics. The authors should consider elaborating in the previous sections on the analysis they performed using these metrics.

- It seems like the kzFO is introduced but barely discussed, and should probably be dropped from the main text.

- The lack of clarity in the figures is mirrored by the introduction of a large number of confusing acronyms in the text (mme_G, mmeS_GR, mmeW_R, kzFO. . .). These are often unnecessary, and the manuscript would greatly benefit from the use of more complete descriptions of the ensembles being discussed even if it means a small increase in length. I would suggest using the following names in place of the acronyms: o mme_GR: Hybrid ensemble o mme_G: Global ensemble o mme_R: Regional ensemble o mmeS_GR: Optimized hybrid ensemble o mmeS_G: Optimized global ensemble o mmeS_R: Optimized regional ensemble o mmeW_GR: Weighted hybrid ensemble o mmeW_G: Weighted global ensemble o mmeW_R: Weighted regional ensemble Furthermore the "kzFO" ensemble is almost never discussed, is not found to offer an improvement over the other ensemble options, and simply adds to the confusion. I would recommend moving all discussion of the kzFO ensemble to the SI.

- Although I understand that this is an ensemble of opportunity and that the authors have no control over what data is available to them, it seems odd to classify hemispheric CMAQ as a "global model".

- The authors should justify their choice not to include urban monitor data, as better

capturing the role of non-linear chemistry in urban environments is stated advantage of regional-scale modeling.

- The fact that different meteorological fields are being used for the different models should be explicitly mentioned by the authors as this could be a key factor in the differences between the various models.

- Two of the global models seem to be nearly identical (models 1 and 2), providing only different resolutions. Can these really be considered as giving "original and independent contributions" (1st paragraph)?

- The information in table 2 is inconsistent – sometimes degree symbols are used, sometimes they aren't; sometimes pressure units are hPa, sometimes mbar. The table would benefit from being cleaned up.

- In table 2, the GEOS-Chem model top should be 0.01 hPa and not 0.066 hPa as listed (0.066 hPa is the second-to-last pressure edge: http://wiki.seas.harvard.edu/geos-chem/index.php/GEOS-Chem_vertical_grids#47-layer_reduced_vertical_grid). This is simply the model I am most familiar with; I recommend that the authors also re-check the details of the other models in both tables.

---

## Author Comment (AC1) · 19 Apr 2018

**Reply to review #1 of the paper:**

*Two-scale multi-model ensemble: Is a hybrid ensemble of opportunity telling us more?"*

by Stefano Galmarini et al.

Let us thank the reviewer for his understanding of the specific relevance of our work and his comments that indeed have improved the manuscript.

Herewith we will respond in a point-by-point fashion, direct responses are in red.

- The paper structure: The paper has been restructured following the reviewer suggestions, which makes it more coherent and readable. The figures have not been reduced in number but some have been included in the supplemental material. The figure organization has been changed slightly.

- Figure labeling and readability: This aspect has also been improved by making the figure more self explanatory.

- Quantitative conclusions summary in the abstract: More care has been given to the abstract an to adding more quantitative conclusions

- Additional points

- In the conclusions the authors refer to analysis of annual hourly, JJA hourly, and annual daily maximum records. However, until that point there seems to be no discussion of the latter two metrics. The authors should consider elaborating in the previous sections on the analysis they performed using these metrics.

It is not clear to us why the reviewer refers to data frequencies as metrics. The annual and JJA time periods are analysed to consider the all period of the simulation and the period in which ozone production is dominant. The daily maximum is a standard indicator of ozone presence. The text has been modified to make these distinctions clearer.

- It seems like the kzFO is introduced but barely discussed, and should probably be dropped from the main text.

kzFO is an ensemble analysis developed in the past that completes the spectrum of the available treatments. It appears to be neglected because it does not produce any substantial improvement. We'd rather keep it as part of the analysis for the sake of completeness. However we have made clear in the text that due to the above mentioned reasons it won't be much analysed.

- The lack of clarity in the figures is mirrored by the introduction of a large number of confusing acronyms in the text (mme_G, mmeS_GR, mmeW_R, kzFO. . .). These are often unnecessary, and the manuscript would greatly benefit from the use of more complete descriptions of the ensembles being discussed even if it means a small increase in length. I would suggest using the following names in place of the acronyms: o mme_GR: Hybrid ensemble o mme_G: Global ensemble o mme_R: Regional ensemble o mmeS_GR: Optimized hybrid ensemble o mmeS_G: Optimized global ensemble o mmeS_R: Optimized regional ensemble o mmeW_GR: Weighted hybrid ensemble o mmeW_G: Weighted global ensemble o mmeW_R: Weighted regional ensemble Furthermore the "kzFO" ensemble is almost never discussed, is not found to offer an improvement over the other ensemble options, and simply adds to the confusion. I would recommend moving all discussion of the kzFO ensemble to the SI.

We agree with the proposed naming strategy the text has been harmonised accordingly

- Although I understand that this is an ensemble of opportunity and that the authors have no control over what data is available to them, it seems odd to classify hemispheric CMAQ as a "global model".

The reviewer is right in the sense that H-CMAQ is a hemispheric model, however for practical purposes, we do not think this distinction makes any difference for the analysis – there is little cross-hemispheric transport on the time scales that were simulated here and the area of analysis is EU only.  We added a statement in the database section that one of the models (H-CMAQ) is not truly global but for the purposes of this analysis is expected to behave the same as global models over the Northern Hemisphere and, therefore, for the rest of the paper we will refer to as "global models. A footnote to the table has been added where it is clear that we acknowledge the fact that H-CMAQ is not strictly global.

- The authors should justify their choice not to include urban monitor data, as better capturing the role of non-linear chemistry in urban environments is stated advantage of regional-scale modeling.

The large difference in model resolution between the global and regional scale models already produces as a result that many more monitoring stations fall in one grid cell of a global model that in a regional one. This produces as an effect that one model value is associated to many monitoring points where high local variability is also measured. Using rural stations, which are in principle less prone to local emissions, should reduce a physiological difference that would only penalise the global models. Based on this argument, it is clear that by adding the large number of urban/sub-urban monitoring station would exacerbate this problem. This point has been clarified in the text.

- The fact that different meteorological fields are being used for the different models should be explicitly mentioned by the authors as this could be a key factor in the differences between the various models.

This point is not hidden but mentioned in section and detailed in the table

- Two of the global models seem to be nearly identical (models 1 and 2), providing only different resolutions. Can these really be considered as giving "original and independent contributions" (1st paragraph)?

Model independence is a nasty beast. Are the other models more independent than these two just because they have different names but may share 75% of modules or methodologies? Is a different name or different resolution sufficient for us to define two models as independent? Obviously not. The one pointed out by the reviewer is a patent case of dependency of which we are aware, it should be pointed out that two of the ensemble methodologies adopted (S and W) are accounting for redundancy and the lack of independence is dealt with in the paper. However excluding 2 models simply because they are two versions of the same one would imply that we can assume that the others are independent when we all know they are not and as the regional scale model redundancy analysis demonstrates.

- The information in table 2 is inconsistent – sometimes degree symbols are used, sometimes they aren't; sometimes pressure units are hPa, sometimes mbar. The table would benefit from being cleaned up.

Units have been harmonised thanks.

- In table 2, the GEOS-Chem model top should be 0.01 hPa and not 0.066 hPa as listed (0.066 hPa is the second-to-last pressure edge: http://wiki.seas.harvard.edu/geoschem/index.php/GEOS-Chem_vertical_grids#47-layer_reduced_vertical_grid). This is simply the model I am most familiar with; I recommend that the authors also re-check the details of the other models in both tables.

We have contacted the model user. He is in agreement with what stated by the reviewer however since the values extracted are generated at the base of the cell, he preferred to give the pressure level of that element which constitutes in fact the 47[th] level rather than the cell top (48[th] level as stated in the GEOSS-CHEM specs). You were both right. Thanks

---

## Author Comment (AC2) · 19 Apr 2018

**Reply to review #2 of the paper:**

*Two-scale multi-model ensemble: Is a hybrid ensemble of opportunity telling us more?"*

by Stefano Galmarini et al.

Let us thank the reviewer for his understanding of the specific relevance of our work and his comments that indeed have improved the manuscript.

It should be noted that surface ozone concentrations (and their observations) are a notoriously difficult measure to analyse and compare against models due to the strong effects of local emissions and deposition, so care should be taken to limit the scope of this manuscript to the performance of the ensemble in terms of surface ozone. This should be discussed in the introduction.

We feel like disagreeing on this, not strictly on the fact that ozone is not prone to local emissions but on the fact that ozone measurements are "notoriously difficult measure to analyse and compare against models". Ozone measurements are accurate, what might be in accurate are the emission inventories used by models. We have taken into account these elements by using only rural stations, which should be far from local sources of ozone precursors and have a larger spatial representativeness.

Before going into details I would finally urge the authors to reconsider putting 14 (!) figures into this manuscript and rather move several of them to a supplement in order to improve readability.

This aspect has been dealt with also in agreement with the requests of rev#1

Specific comments:

P2, L70 define which "spectra" you refer to here.

thanks

P2, L74-75 is it a good thing that you use reg-glob equally at only 15 % of stations?

It is a fact with no qualification attached. It is clear that the larger the number of stations where both model type contribute the clearer is the level of complementarity of the set. A clarification of this statement has been added to the text

P5, L60-66 this is a slightly confusing amalgamation of arguments. Limitations in space are combined with different representations of gas-phase chemistry. I suggest

rephrasing this paragraph and simply discussing gas-phase chemistry mechanism diversity.

A clarification of this statement has been added to the text. We left the difference in domain pertinence of the two groups which is connected to the time scale of the chemistry represented.

P7, L221: regional models biased towards C-IFS?

Well in principle yes as far as boundary conditions are concerned but we do not see this in the results.

P7, L229: "can take stock" seems inappropriately used in this context, revisit expression. Figure 2: please use the same axis (Period in days is fine) for 2a as for 2b and c.

Correct, thanks

P8, L249ff: why did you average the spectrum? how did you average the spectrum? Are 24h peaks as pronounced in the model as in the measurements?

The spectrum is smoothed with a Savitzky-Golay filter to allow overlying of multiple spectra for comparison. In the new Figure 1 (ex Figure 2), we did not average in order to make the most characteristic peaks visible.

P8, L249ff: Spectrum analysis also results in mangling daily maxima and daily minima. I am not sure if I like the fact that this means you are evaluating both photochemistry (daily maxima) and representation of boundary layer parameterization (nighttime titration)... Could be worth discussing in the text.

It is impossible to disentangle the dynamics form the chemistry and represent the process separately. The nighttime titration is also affected by BL dynamics due to the suppression of convection and a shallowing of the BL. But probably we do not get the point of the reviewer here.

P8, L255: "The time series of the rural monitoring ...", these are simply your "observations", right? Not a subset or something? I suggest replacing "rural monitoring stations" with observations so as not to confuse readers.

Monitoring stations of operational networks are individually classified according to the location in which they reside into rural, sub-urban and urban. No arbitrary judgment form our side has interfered with the station classification. But may be we miss interpreted: " "The time series of the rural monitoring ...", these are simply your "observations", right? Not a subset or something''

P8, L268: Interesting! Do you dare to speculate as to why this is?

The models show generally higher variability and energy in the inter-diurnal range while the two spectrums are at good agreement at mesoscale and synoptic scales. Those facts are directly related to the resolution of the models and hence the resolved features.

P8, L269-271: "A weak sec..." this sentence seems repetitive of what comes afterwards, remove or merge with remainder of paragraph.

Sorry your comment is not clear, the analysis is performed per period range and there is no repetition in the ranges analysis.

P9, L298-300: "An element of surprise...". Surprising sentence - it reads like the introduction to the paragraph before. Why is it at the end? Is it a conclusion? Might want to rephrase.

Corrected

P11, L337-338: if it is important, please tell us what you think about GLO vs REG now. Is the fact that GLO have a higher POD but also a higher FAR a good thing? Not really, no?

We present a multi-parameter evaluation of global and regional models. There could be no good or bad in a "global" sense, neither that is the aim of this exercise.

P12, L371: "... transport in the case of a global model, " are you not talking about regional models here? Consider cleaning up the paragraph.

Corrected

P13, L401ff: I disagree with this assessment of the combined histogram (Talagrand diagram). The regional models (6b) actually came quite close to the ideal Talagrand diagram (equal distribution). Combining them and increasing bin number does not (necessarily) increase the value range. Now you still have an overdispersed model system, which just happens to be more correctly distributed amongst values closer to the mean (bins 5-18). I suggest to rephrase this paragraph.

The sentence has been rephrased, thanks

Fig 9: very small panels, please improve this to make it readable. C3 ACPD Interactive comment Printer-friendly version Discussion paper

Corrected

P17, L532: Figure 11 does not have panels a, b and c...

Corrected, thanks

P17, L539ff: calling this improvement "systematic" is an exaggeration - there is no estimate of the uncertainty of these numbers, hence you have no idea whether the difference ("improvement") is statistically significant. This should be rephrased and written more cautious.

It is systematic through out the case analysed, not in a statistical sense.

Fig 13, and also P18, L572-573: it would be very helpful if the optimal point would be marked in these plots. If I understand this plot correctly, for a 6 member ensemble the optimal point is (x=16,667, y=1; description in text says in line 573: (x=100/(Number of Models),y=1) with number of models = 6). All points are pretty far off of this optimal point in all the plots!

The optimal point corresponds to the i.i.d. situation, i.e. an ensemble with independent and identically distributed (around the truth) models.

P18, L579: typo "form"

Corrected thanks

P18, L580ff: y scales 0-1, hence the "y values" cited here are for x, probably.

Corrected thanks

P18, L583: I guess you are talking about Figure 13d (mmeS6) as being superior here - rephrase to make this clear.

Corrected thanks

Fig 14: legends miss "obs" description (if that is what the thick red line is supposed to show)

Figure changed

P19, 587-600: this paragraph deserves its own (sub)caption, as it concerns a quite different and important point than the previous text

Not clear what a sub-caption of a paragraph is, sorry.